# Hyperosmolar Treatment for Patients at Risk for Increased Intracranial Pressure: A Single-Center Cohort Study

**DOI:** 10.3390/ijerph17124573

**Published:** 2020-06-25

**Authors:** Agnieszka Wiórek, Tomasz Jaworski, Łukasz J. Krzych

**Affiliations:** Department of Anesthesiology and Intensive Care, Faculty of Medical Sciences in Katowice, Medical University of Silesia, 14 Medyków Street, 40–752 Katowice, Poland; tomasz.jaworski@mp.pl

**Keywords:** hyperosmolar therapy, intracranial pressure, intracranial hypertension, hypertonic saline, mannitol treatment

## Abstract

Treatment with osmoactive agents such as mannitol or hypertonic saline (HTS) solutions is widely used to manage or prevent the increase of intracranial pressure (ICP) in central nervous system (CNS) disorders. We sought to evaluate the variability and mean plasma concentrations of the water and electrolyte balance parameters in critically ill patients treated with osmotic therapy and their influence on mortality. This cohort study covered patients hospitalized in an intensive care unit (ICU) from January 2017 to June 2019 with presumed increased ICP or considered to be at risk of it, treated with 15% mannitol (G1, n = 27), a combination of 15% mannitol and 10% hypertonic saline (HTS) (G2, n = 33) or 10% HTS only (G3, n = 13). Coefficients of variation (Cv) and arithmetic means (mean) were calculated for the parameters reflecting the water and electrolyte balance, i.e., sodium (NaCv/NaMean), chloride (ClCv/ClMean) and osmolality (mOsmCv/mOsmMean). In-hospital mortality was also analyzed. The study group comprised 73 individuals (36 men, 49%). Mortality was 67% (n = 49). Median NaCv (G1: *p* = 0.002, G3: *p* = 0.03), ClCv (G1: *p* = 0.02, G3: *p* = 0.04) and mOsmCv (G1: *p* = 0.001, G3: *p* = 0.02) were higher in deceased patients. NaMean (*p* = 0.004), ClMean (*p* = 0.04), mOsmMean (*p* = 0.003) were higher in deceased patients in G3. In G1: NaCv (AUC = 0.929, *p* < 0.0001), ClCv (AUC = 0.817, *p* = 0.0005), mOsmCv (AUC = 0.937, *p* < 0.0001) and in G3: NaMean (AUC = 0.976, *p* < 0.001), mOsmCv (AUC = 0.881, *p* = 0.002), mOsmMean (AUC = 1.00, *p* < 0.001) were the best predictors of mortality. The overall mortality prediction for combined G1+G2+G3 was very good, with AUC = 0.886 (*p* = 0.0002). The mortality of critically ill patients treated with osmotic agents is high. Electrolyte disequilibrium is the independent predictor of mortality regardless of the treatment method used. Variations of plasma sodium, chloride and osmolality are the most deleterious factors regardless of the absolute values of these parameters

## 1. Introduction

Intracranial hypertension (ICH) appears as a severe complication of many disorders of the central nervous system (CNS), such as non-traumatic and traumatic brain injury and many others [1]. It is usually defined as a sustained elevation of intracranial pressure (ICP) above 20 mmHg [2]. What makes the condition fatal is not always the primary insult to the brain tissue, but the development of cerebral edema in the following days [3]. Treatment of elevated ICP includes surgical and conservative methods of reducing the volume of the cerebral tissue, the latter based mainly on osmotic agents and hyperosmolar solutions [4]. Though plenty of studies regarding the efficiency of osmotherapeutics exist, there is still insufficient discussion concerning the clinical circumstances in which the application of osmotherapy leads to the improvement of patients’ outcomes. Many recent studies suggest putting surgical methods, such as decompressive craniectomy (DC) at the forefront of investigation as perhaps superior in the treatment of high ICP [5]. This superiority is most noticeable when referring to the cytotoxic type of cerebral edema, developing most often in the course of traumatic brain injury or ischemic stroke.

Treatment with osmoactive agents is widely used and often highly praised. However, there are major differences between practices implemented in various centers and intensive care units. Mannitol was the main focus of research since the 1970s and 1980s and has remained the first-line therapy across the United Kingdom and many regions of the USA, as well as in Poland and Germany, whereas hypertonic saline solutions have been properly explored just over the past 15 years and are currently gaining popularity worldwide [6,7,8]. Studies referring to “hypertonic” saline show the adaptation of concentrations from 3% to even 23.5% sodium chloride [9]. Discrepancies concern also whether the osmotherapeutics should be used as boluses, in continuous infusions or as a mixed regimen, as well as what values of sodium and osmolality should be achieved and not exceeded (sources most often refer to a sodium value of 145 to 155-mmol/L and an osmolality value of 310 to 320 mOsmol/kg [10,11]. In addition, there is no agreement what variations of those parameters during therapy can be considered “safe”.

Many methods of cerebral edema treatment exist that include various practices within osmotherapy. This makes it difficult to compare the results of different studies. It is also difficult to draw binding conclusions regarding the efficacy and drawbacks of implementing this treatment.

Taking into account the above mentioned rationale, we sought to: (1) to investigate the mean plasma concentrations and the variability of selected arterial-blood gas parameters in critically ill patients treated with osmoactive agents, and; (2) to investigate the association between the mean plasma concentrations and their observed variability and the patient’s outcome.

## 2. Materials and Methods

In this observational cohort study, we included consecutive patients who were admitted to the intensive care unit (ICU) between 1 January 2017 and 30 June 2019 in the Department of Anesthesiology and Intensive Care at the hospital of the Medical University of Silesia in Katowice, Poland. The patients were admitted to the ICU with presumed increased ICP or considered to be at risk of it, regardless of its origin. The underlying causes of ICU admissions and treatment were traumatic or non-traumatic brain injuries with a subsequent cerebral edema. The traumatic brain injuries (TBI) were as follows: TBI with intracerebral hemorrhage (n = 11/15%); TBI with subdural/epidural hematoma (n = 14/19%); and diffuse TBI (n = 13/18%). The non-traumatic brain injuries were as follows: subarachnoid hemorrhage (SAH) caused by a ruptured aneurysm (n = 19/26%); neoplasms (n = 4/5.5%); and stroke/cerebral infarction (n = 12/16.5%).

The study population included patients qualified for the implementation of osmotic therapy as a means to manage or prevent the development of increased ICP. No a priori power or sample calculation was performed. The selection process of the participants is depicted in Figure 1.

Demographic and medical data were recorded based on the proprietary medical documentation, including sex, age, APACHE II score and length of ICU stay. Under Sections 21 and 22 of the Polish Law of 5 December 1996 concerning the medical profession [12] and due to the non-interventional design of the study, the ethics committee waivered the need for obtaining approval (PCN/0022/KB/258/I/19). However, all patients’ data were obtained in accordance with the Polish legal regulations concerning personal data management, after the written consent was given by the patients on hospital admission, excluding unconscious patients who required emergency procedures.

Based on the initial cause of ICH, the patients were treated with a 15% mannitol solution, 10% hypertonic saline (NaCl) or the combination of both solutions. Patients were divided into three groups based on the treatment received, namely: Group 1 (G1, n = 27/37%) treated solely with mannitol solution during their entire ICU stay; Group 2 (G2, n = 33/45%) where patients received both 15% mannitol and 10% NaCl solution adequate to their clinical condition; and Group 3 (G3, n = 13/18%) where patients received 10% hypertonic saline through the course of treatment, due to contraindications for mannitol administration.

The treatment applied to all selected patients was in agreement with the hospital’s standard operating procedure (SOP) containing evidence-based medicine guidelines for managing ICH. The SOP includes the use of 10% hypertonic saline administered as an infusion of 50 mL of the solution for 30 min every 6 h or a continuous infusion of 8 mL/h in patients with TBI, ICH, SAH. In other cases, the standard procedure instructs one to administer 100 mL of 15% mannitol solution intravenously in 30-min-long infusion every 6 h. The target Na+ concentration observed in arterial blood gases analysis should not exceed 155 mmol/L with a target plasma osmolality of no more than 320 mOsm/kg. The decision regarding the course of treatment chosen for a specific patient (i.e., patient-oriented treatment) was made by the attending physicians based on the best of their knowledge and the patient’s overall condition and was not in any way influenced by a third party. The treatment was personalized, taking into consideration electrolyte balance and plasma osmolality.

The patients had their optic nerve sheath diameter (ONSD) measured on a daily basis and the osmotic treatment was continued until brain edema resolution, as assessed by the clinical status and ONSD or death resulted. Shortly, the ultrasonic measurement of the optic nerve sheath diameter is a non-invasive method for ICP monitoring. An increase in ICP in the intracranial compartment is reflected by swelling of the optic nerve and, as a result, an increase in the diameter of the optic nerve sheath. The advantages of the method come from the fact as that it is non-invasive, produces quick results, does not require sophisticated equipment, and can be performed at the bedside, there is no need to transport critically ill patients outside of the ICU [13]. The ONSD measurement technique was extensively studied while the principles behind it, as well as any factors that could impact on it, have been previously described [14,15]. There is an ongoing issue, however, regarding which ultrasonography mode is best for evaluating the ocular structures. This was discussed, for example, by De Bernardo et al. who suggested using a one-dimensional amplitude-mode (A-scan) technique over a two-dimensional brightness-mode (B-scan) [16]. However, a two-dimensional B-mode is also an accepted technique verified in transorbital ultrasonography according to available systematic reviews and a meta-analysis [17,18]. In our study, the ONSD was measured bilaterally every 24 h in the late afternoon by two medical personnel skilled in the technique. The measurements were performed with the eyelids closed, with the patient in a supine position, using a two-dimensional mode with a linear (14.0–6.0 MHz) ultrasound transducer (M7, Mindray, Shenzhen, China), with the caliper being set at 3 mm posteriorly to the papilla. To minimize error, we calculated the mean from three measurement results in each eye. We paid great attention to avoid excessive pressure on the eyeball. To prevent any potential damage to the ocular structures, we complied with the ALARA principle (as low as reasonably achievable), according to the British Medical Ultrasound Society [19,20]. We set the upper limit for the ONSD at 5.7 mm [21]. In our ICU, ultrasonic measurement of the ONSD is routinely used for monitoring patients at risk of an increased ICP, particularly when sedation hinders neurological examination [22].

We performed arterial blood gas (ABG) tests on all patients twice a day at 12-h intervals and assessed them with a Siemens RAPIDPoint 500 blood gas system analyzer (Erlangen, Germany). Absolute values of plasma sodium (Na), chloride (Cl) and osmolality (Osm) were recorded and the values of the strong ion difference (SID) were calculated based on the accepted equation (SID = [Na^+^+K^+^]-[Cl^−^+lactates]) [23]. Coefficients of variation (Cv, [%]) were calculated in order to express the variability of the selected ABG parameters reflecting the water and electrolyte balance, namely: sodium (NaCv); chloride (ClCv); plasma osmolality (mOsmCv); and the strong ion difference (SIDCv). Cvs were calculated based on the equation Cv = S/X*100%, where “S” stands for standard deviation and “X” stands for the arithmetic mean of all measurements taken. The arithmetic means of all measurements taken during treatment were also calculated for the ABG parameters of interest, namely: sodium (NaMean); chloride (ClMean); plasma osmolality (mOsmMean); and the strong ion difference (SIDMean).

Patients were followed-up until ICU discharge or death. ICU mortality (i.e., the number of deaths during the index hospitalization) was considered as the outcome.

The STROBE Statement (strengthening the reporting of observational studies in epidemiology) was applied for appropriate data reporting.

A statistical analysis was performed using MedCalc v.18 software (MedCalc Software, Ostend, Belgium). Quantitative variables were depicted using medians and interquartile ranges (IQR, i.e., 25pc–75pc). The Shapiro–Wilk test was used to verify their distributions. Qualitative variables were described using frequencies and percentages. Between-group differences for continuous variables were assessed using the Kruskal–Wallis test, while for categorical variables the chi-squared test was applied. The values of the coefficients of variation and arithmetic means were divided into four equal numerical intervals based on quartile values.

Logistic regression was performed in order to verify observations from bivariate models. Variables with a ‘*p*’ value <0.1 in bivariate comparisons were consecutively subjected to a multivariable analysis with the logistic regression model. NaV, ClV, mOsmV were considered the exposure variables, demographic (age, gender) and ICU-stay-related data (ICU length of stay, duration of osmotherapy) were the confounding variables, while the outcome variable was mortality. Logistic ORs (95% CI) were subsequently estimated. ROC curves were drawn and areas under the ROC curves (AUC) were calculated in order to determine the predictive value of each studied parameter and the outcome. Parameters were used individually as the input in order to create the ROC curves in three subgroups. AUCs were also calculated in order to assess the diagnostic accuracy of the final logistic regression equations.

All tests were two-sided. A *p*-value of <0.05 was considered significant.

## 3. Results

The study group consisted of 73 patients (36 men, 49%). The median age was 56 years (IQR 40–68). The median length of the ICU stay was 9 days (IQR 5–15). Detailed characteristics of the study subgroups are depicted in Table 1. The highest APACHE II score was found in patients who were treated both with mannitol and a 10% NaCl solution.

The medians of coefficients of variation for the study population were as follows: NaCv 3.24% (IQR 2.37–4.76); ClCv 4.37% (IQR 3.58–6.45); and mOsmCv 3.39% (IQR 2.32–4.98), respectively. The medians of the arithmetic means of studied parameters throughout the treatment period were as follows: NaMean 142.8 mmol/L (IQR 138.58–149); ClMean 111.0 (IQR 106–118) and mOsmMean 294.60-mmol/L (IQR 285.23–306.38), respectively. Median values of the analyzed parameters in consecutive subgroups are depicted in Table 2. NaMean was highest in patients treated with 10% NaCl and lowest in the mannitol group.

All the investigated parameters did not differ in terms of gender and did not correlate significantly with age or duration of ICU stay (*p* > 0.05 for all, data not shown).

Mortality was 67% (n = 49). The results of bivariate investigations performed within three subgroups divided in terms of mortality are depicted in Table 3. All, but one investigated parameter (i.e., SID) were statistically significantly higher in deceased patients treated with 10% NaCl. Significantly higher variations of Na, Cl, osmolality and SID were also noticed among deceased subjects who was treated with mannitol.

Mortality prediction by the variability and means of all parameters in the ROC curve analysis is depicted in Table 4. It was confirmed that high values of Na, Cl and osmolality predicted the outcome better in patients treated with 10% NaCl, whereas high variations in these parameters were more powerful in subjects treated with mannitol.

Among patients who were treated with mannitol only (G1), those who reached at least 2.65% of NaCV, 4.81% of ClCV or 2.95% of mOsmCV, died. Among subjects treated with 10% NaCl (G3), those who reached at least 150 mmol/L of NaMean, 118 mmol/L of ClMean or 305 mmol/L of mOsmMean, died.

In a logistic regression, it was confirmed that even after adjustment for the patient- and procedure-related variables, mortality was dependent on NaMean, NaCv, mOsmMean and mOsmCv (Table 5). The overall diagnostic accuracy was very good, with AUC = 0.886 (*p* = 0.0002).

## 4. Discussion

This study attempted to explore certain doubts regarding the current trends and utility of hyperosmolar therapy through observational analysis of cases with ICH or those considered to be at risk of it. Most patients suffered from cerebral edema as a result of traumatic or non-traumatic brain injury. We assumed that the water and electrolyte balance may dangerously change during treatment with different osmotic agents, with a deleterious impact on the outcome.

Cerebral edema develops from several pathologic mechanisms arising as a consequence of primary and secondary injury to the brain [24]. A displacement of the brain tissue occurs with an increase of ICP, leading to cerebral herniation and a rapid worsening of prognosis [25,26]. The means to alleviate intracranial hypertension and reduce further adverse outcomes have been extensively studied [26,27,28]. Nevertheless, according to the current edition of Brain Trauma Foundation guidelines, none of the universally utilized methods has presented irrefutable evidence of reliable outcome improvement [29]. Since the availability of neurosurgical interventions in most medical facilities is limited, the means of relieving excessive volume accumulated in the brain cells by inducing a hyperosmolar environment are currently considered as a gold-standard treatment [30]. In recent years, most studies have revolved around the question of what combination of medications is the best version of osmotic therapy. Rarely has the question been posed whether osmoactive drugs should even be used in the first place. Even though osmotic therapy has been found efficient in past studies, the lack of definite ramifications and instructions on the issue allows for quite an amount of freedom in deciding what drugs to use or how to dose them, thereby causing disagreements among clinicians and making the results acquired in various institutions difficult to compare objectively [31].

Moreover, voices criticizing the routine use of osmotic therapy are not a novel development in the field. In their respective studies, Mortazavi et al. and Grande et al. referred to the potential occurrence of the rebound phenomenon when pointing out weaknesses and side effects of osmotherapeutics [32,33]. This results from the transiency of cell shrinkage, volume reduction and hemodynamic effects gained from the infusion of osmotic agents [34]. White et al. showed the association between the repeated dosage of osmoactive agents and an increase in ICP and edema up to the values nearing or even exceeding, the initial pre-treatment measurements [35]. When the state of hyperosmolarity persists in the CNS, the idiogenic intracellular osmoles rise, drawing water back to the brain cells. After the cells reach their baseline size and both intra- and extracellular compartments become hyperosmolar, volume reduction of the brain no longer takes place [36]. Diringer, in turn, connected the risk of a rebound edema effect to excessively rapid correction of hyperosmolarity [37]. He showed that intermittent and sudden changes in osmolarity, reflected in the high lability of values before returning to the reference level, may provoke unfavorable consequences. Hyperosmolarity was also connected with increased mortality by Shen et al. in critically ill patients with not only cerebral causes of hospital admission, but also presenting cardiovascular and gastrointestinal disorders, with a threshold osmolarity of 300 mOsm/L [38]. These observations are in line with our findings, as the variability of osmolality was a good predictor of mortality in two studied subgroups with significantly higher CVs among deceased patients. Noticeably high variability and mean values of plasma sodium and osmolality were also an good overall predictor of mortality in the multivariate analysis for the entire studied population. Similar observations have been made in the past, adding to the lack of evidence persisting over the years concerning the actual influence of osmotherapeutics on improved outcome [39]. Although the passage of time and further studies continue to contribute to our knowledge base, remaining doubts and calls for multicenter RCTs regarding the use of osmotherapy and outcome prediction may also be observed in relatively recent publications [40,41].

This inability to draw firm conclusions may come from variability in the effectiveness of applied osmotherapeutics, depending on CNS and cerebral tissue stability. The key aspect is the state of preservation or disruption of the blood–brain barrier (BBB). Both mannitol and hypertonic saline have a high osmotic reflection coefficient, which means the BBB with uninterrupted integrity is highly impervious to their particles [6]. However, in the case of a cerebral tissue disorder, the integrity of the BBB may be compromised, leading to increased permeability to solutes and intensified hydraulic conductivity [42].

A high variability in sodium plasma concentration and plasma osmolality has been connected to neurologic complications, such as encephalopathy, lethargy, seizures and even coma. Shrinkage of cerebral cells caused by a state of hyperosmolarity causes a retraction of the tissue away from the protective dura, which may lead to rupture of the bridging veins and subdural hematomas [43]. The potential of each individual osmoactive agent to cause adverse effects, such as hemodilution, hemolysis, hypotension, electrolyte disturbances, serious renal injury, hyperchloremic acidosis, pulmonary edema, must also not be ignored [33].

Although the effect on ICP decrease by the osmoactive agents is visible, the consequences of a persistent dosage of osmotherapeutics, via repeated boluses or continuous infusions suggest using them rather as a temporary bridge, maintaining cerebral perfusion pressure, preventing cerebral herniation syndrome until other diagnostic and therapeutic procedures can be implemented, such as evacuation of a hematoma, lesions or a decompressive craniectomy [33].

A recent randomized controlled trial aimed at evaluating whether the performance of a decompressive craniectomy (DC) exerts a decisive impact on clinical outcomes in patients with intracranial hypertension, especially as a secondary intervention after ICP-targeted preservative treatment has failed [44]. It concluded that a DC for severe and refractory ICH produced a survival advantage in dependent and independent living alike compared with medical management alone, perhaps due to improved ICP control with surgery. As there was, however, a higher rate of vegetative states and lower or upper disability in the DC-subjected group, the field remains open to further investigation.

### Study Limitations

One should bear in mind the potential limitations of our study, including the relatively small study group. As the study is observational in design, an interventional investigation should be performed to confirm our findings. In addition, as this a single-center observation, extrapolation of our results is limited. No power analysis was performed to assess the sample size, introducing a potential selection bias. Although the treatment applied was approved by the hospital’s SOP, the final osmotherapy regimen was decided subjectively by the patient’s attending physician, creating another possibility of a bias. However, out of 1212 examined cases, only the 73 chosen presented indications for osmotic therapy. Finally, the diagnosis of ICH was primarily based on a clinical assessment while the ultrasonic measurement of the optic nerve sheath diameter was the only method routinely used in our patients. Therefore, we did not observe a direct effect of the osmotic therapy in the decrease of ICP. However, as mentioned in the methodology of the study, the treatment was continued until we observed an improvement indicating brain edema regression as assessed by the clinical status and ONSD. Although ONSD is a non-invasive method of ICP monitoring which creates a possibility of a false measurement, in our study all ONSD measurements were taken by at least two trained specialists with experience of using this method in clinical practice. The technique of a B-scan two-dimensional ultrasonography used in our study could be seen as a limitation of our study, due to a so-called “blooming-effect” [16]. However, there is enough published evidence showing the efficacy of the B-scan and there are analyses of the improvement and the development of higher frequency probes allowing brightness mode (B-mode) scanning to replace A-mode sonography [45].

## 5. Conclusions

The mortality of critically ill patients treated with osmotic agents is high. Electrolyte disequilibrium is the independent predictor of mortality regardless of the treatment method used. Variations of sodium, chloride and osmolality are the most deleterious factors involved regardless of the absolute values of these parameters. Further research is needed to set secure boundaries of osmotic therapy in the context of changes of the water–electrolyte balance parameters.

## Figures and Tables

**Figure 1 ijerph-17-04573-f001:**
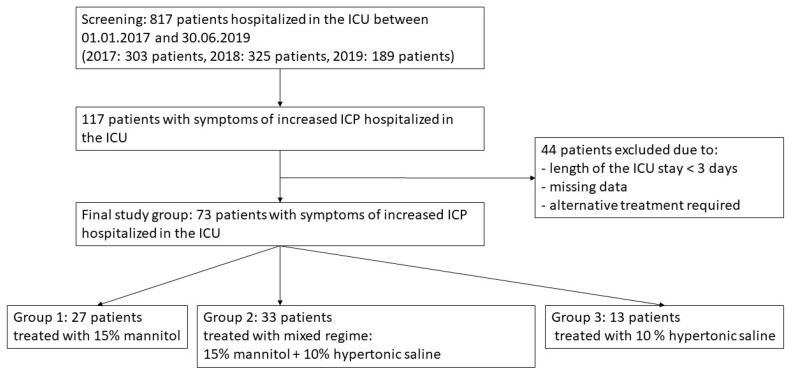
Flowchart of the study population.

**Table 1 ijerph-17-04573-t001:** Study group characteristics and procedure-related variables ^1^.

Variable	Unit	G1 (n = 27)	G2 (n = 33)	G3 (n = 13)	‘*p*’
Sex	Males	14 (52%)	16 (48%)	6 (46%)	0.9
	Females	13 (48%)	17 (52%)	7 (54%)	
Age	(years)	52 (35–66)	65 (51.5–70)	50 (42–57)	0.4
Duration of the therapy	(days)	5 (3–14)	6 (2–12)	3 (2.75–4.25)	0.06
Total dose of mannitol	(grams)	30 (22.8–42.4)	20 (13.8–30.9)	N/A	–
APACHE II score	(points)	17 (15.0–23.0)	19.5 (15.5–26)	16 (12–18.5)	0.03
Mortality	Death	21 (78%)	22 (67%)	6 (46%)	0.1
	Survival	6 (22%)	11 (33%)	7 (54%)	

^1^ Values are medians, interquartile ranges (Q1–Q3) for quantitative variables and frequencies and percentages for qualitative variables. APACHE II score, IQR—interquartile range, N/A—not applicable.

**Table 2 ijerph-17-04573-t002:** Water and electrolyte balance during treatment with osmotic agents ^2^.

Variable	G1 (n = 27)	G2 (n = 33)	G3 (n = 13)	‘*p*’
NaCv	2.86 (2.35–4.61)	3.31 (2.44–4.59)	3.88 (2.32–6.32)	0.7
NaMean	140.9 (137.3–148.1)	142.6 (138.6–146.9)	147.8 (144.1–151.6)	0.03
ClCv	4.35 (3.15–6.21)	4.37 (3.72–6.45)	4.45 (3.56–7.61)	0.7
ClMean	109 (105–118)	110 (105–116)	115 (112–120)	0.09
mOsmCv	3.29 (2.32–5.22)	3.39 (2.32–4.47)	3.89 (2.27–6.28)	0.8
mOsmMean	293.6 (282.0–306.7)	292.8 (286.0–301.4)	302.8 (295.4–311.2)	0.07
SIDCv	10.93 (8.33–13.99)	10.25 (7.68–12.58)	10.19 (8.83–14.39)	0.7
SIDMean	32.72 (30.82–34.24)	34.08 (33.55–35.68)	33.95 (32.14–35.30)	0.06

^2^ Values are medians and IQR. ’*p*’ values refer to differences in strata defined by consecutive variables (e.g., NaCv, NaMean, etc.). NaCv—coefficient of variation for plasma sodium concentration (plasma sodium variability); NaMean—mean value of plasma sodium concentration; ClCv—coefficient of variation for plasma chloride concentration (plasma chloride variability); ClMean—mean value of plasma chloride concentration; mOsmCv—coefficient of variation for plasma osmolality (plasma osmolality variability); mOsmMean—mean value of plasma osmolality; SIDCv—coefficient of variation for strong ion difference (strong ion difference variability); SIDMean—mean value of strong ion difference.

**Table 3 ijerph-17-04573-t003:** Water and electrolyte balance during treatment with osmotic agents and their association with mortality ^3^.

Variable	G1 (n = 27)			G2 (n = 33)			G3 (n = 13)		
	Survival n = 6	Death n = 21	‘*p*’	Survival n = 11	Death n = 22	‘*p*’	Survival n = 7	Death n = 6	‘*p*’
NaCv	2.07 (1.51–2.35)	3.74 (2.76–5.32)	0.002	3.90 (2.43–5.04)	3.30 (2.71–3.84)	0.6	2.74 (2.07–3.72)	6.40 (5.91–6.67)	0.03
NaMean	137.8 (136.5–139.1)	142.8 (138.3–149.4)	0.1	139.4 (136.7–142.2)	143.0 (140.2–150.1)	0.06	144.4 (142.3–147.2)	151.9 (150.0–152.9)	0.004
ClCv	3.09 (1.63–3.57)	5.65 (3.30–6.88)	0.02	4.11 (3.74–6.91)	4.45 (3.70–5.99)	0.9	3.80 (3.49–4.43)	8.40 (5.38–10.25)	0.04
ClMean	108 (103–109)	113 (106–122)	0.1	108 (104–111)	112 (106–117)	0.1	113 (112–115)	120 (118–121)	0.04
mOsmCv	2.05 (2.03–2.22)	4.23 (3.03–5.55)	0.001	4.04 (2.38–4.89)	3.39 (2.28–4.03)	0.6	2.40 (2.02–3.72)	6.29 (5.40–6.94)	0.02
mOsmMean	282.3 (279.6–286.4)	295.0 (284.4–309.0)	0.06	286.4 (279.4–291.7)	295.7 (288.6–308.7)	0.03	295.7 (290.3–301.4)	311.3 (309.8–314.0)	0.003
SIDCv	7.69 (7.47–9.90)	11.20 (9.72–15.4)	0.01	10.87 (9.69–12.60)	9.34 (7.13–12.45)	0.2	10.19 (8.38–12.75)	12.47 (9.28–15.59)	0.3
SIDMean	34.88 (32.36–37.79)	32.18 (30.60–34.03)	0.1	34.08 (33.68–34.29)	34.16 (33.27–35.93)	0.7	33.95 (31.94–35.71)	34.36 (32.34–35.03)	1.0

^3^ Values are medians and IQR. ‘*p*’ values refer to differences in strata defined by consecutive variables (e.g., NaCv, NaMean, etc.). NaCv—coefficient of variation for plasma sodium concentration (plasma sodium variability); NaMean—mean value of plasma sodium concentration; ClCv—coefficient of variation for plasma chloride concentration (plasma chloride variability); ClMean—mean value of plasma chloride concentration; mOsmCv—coefficient of variation for plasma osmolality (plasma osmolality variability); mOsmMean—mean value of plasma osmolality; SIDCv—coefficient of variation for strong ion difference (strong ion difference variability); SIDMean—mean value of strong ion difference; IQR—interquartile range.

**Table 4 ijerph-17-04573-t004:** ROC curve analysis for mortality prediction by the investigated parameters within the three subgroups ^4^.

Variable	G1 (n = 27)		G2 (n = 33)		G3 (n = 13)	
	AUC (95% CI)	‘*p*’	AUC (95% CI)	‘*p*’	AUC (95% CI)	‘*p*’
NaCv	0.929 (0.761–0.992)	<0.0001	0.554 (0.371–0.726)	0.6	0.857 (0.558–0.984)	0.01
NaMean	0.722 (0.518–0.876)	0.03	0.705 (0.521–0.850)	0.03	0.976 (0.715–1.000)	<0.001
ClCv	0.817 (0.622–0.939)	0.0005	0.514 (0.335–0.691)	0.9	0.833 (0.531–0.976)	0.04
ClMean	0.718 (0.513–0.873)	0.04	0.661 (0.476–0.816)	0.1	0.845 (0.544–0.980)	0.03
mOsmCv	0.937 (0.772–0.994)	*p* < 0.0001	0.562 (0.379–0.733)	0.6	0.881 (0.587–0.991)	0.002
mOsmMean	0.754 (0.551–0.898)	0.01	0.738 (0.556–0.875)	0.008	1.000 (0.753–1.000)	<0.001
SIDCv	0.849 (0.660–0.957)	<0.0001	0.630 (0.445–0.791)	0.2	0.667 (0.362–0.894)	0.3
SIDMean	0.722 (0.518–0.876)	0.09	0.539 (0.358–0.713)	0.7	0.500 (0.221–0.779)	1.0

^4^ AUC—area under the ROC curve; NaC—coefficient of variation for plasma sodium concentration (plasma sodium variability); NaMean—mean value of plasma sodium concentration; ClCv—coefficient of variation for plasma chloride concentration (plasma chloride variability); ClMean—mean value of plasma chloride concentration; mOsmCv—coefficient of variation for plasma osmolality (plasma osmolality variability); mOsmMean—mean value of plasma osmolality; SIDCv—coefficient of variation for strong ion difference (strong ion difference variability); SIDMean—mean value of strong ion difference.

**Table 5 ijerph-17-04573-t005:** Mortality prediction in multivariate analysis ^5^.

Variable	Mortality Prediction
Sex (Female = 0/Male = 1)	OR = 1.74; 95% CI; 0.47 to 6.43 *p* = 0.4
Age (per 1 year)	OR = 1.01; 95% CI 0.97–1.06; *p* = 0.6
ICU length of stay (per 1 day)	OR = 0.98; 95% CI 0.82–1.17; *p* = 0.8
Duration of osmotherapy (per 1 day)	OR = 0.98; 95% CI 0.83–1.16; *p* = 0.8
NaCv (per 1%)	OR = 0.09; 95% CI 0.01–0.61; *p* = 0.01
NaMean (per 1 mmol/L)	OR = 0.22; 95% CI 0.07–0.66; *p* = 0.007
mOsmCv (per 1%)	OR = 8.22; 95% CI 1.51–44.81; *p* = 0.01
mOsmMean (per 1 mmol/L)	OR = 2.40; 95% CI 1.34–4.29; *p* = 0.003
ClCv (per 1%)	OR = 1.42; 95% CI 0.83–2.42 *p* = 0.2
ClMean (per 1 mmol/L)	OR = 0.85; 95% CI 0.63–1.14 *p* = 0.3
AUC for the final logistic model	0.886; 95% CI 0.790–0.949; *p* = 0.0002

^5^ AUC—area under the ROC curve; NaCv—coefficient of variation for plasma sodium concentration (plasma sodium variability); NaMean—mean value of plasma sodium concentration; ClCv—coefficient of variation for plasma chloride concentration (plasma chloride variability); ClMean—mean value of plasma chloride concentration; mOsmCv—coefficient of variation for plasma osmolality (plasma osmolality variability); mOsmMean—mean value of plasma osmolality.

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
