# Peer review of "Hyperosmolar Treatment for Patients at Risk for Increased Intracranial Pressure: A Single-Center Cohort Study"

_ijerph, 2020, doi:10.3390/ijerph17124573_

Round 1
Reviewer 1 Report
I read this manuscript by Wiorek et al with great interest. Unfortunately, the manuscript is poorly written with lots of grammar errors (see my specific comments). The authors need to find a local English speaker to do proof reading. The authors compared therapy with osmoactive agents i.e. mannitol and hypertonic saline (HTS) solutions for increased intracranial pressure. They concluded that treatment of increased ICP with mannitol or HTS may increase the variability of sodium, chloride and plasma and may increase patient mortality. I have some major concerns as below:
- This paper is related to increased intracranial pressure (ICP), however, the authors did not record ICP values and did not show the effect of the drugs on decreasing ICP;
- They did multiple arterial blood gas tests for each patient, could they give more details? How many tests did they do each day?
- In the logistic regression, what kind of demographic factors were put into the model? Age? Gender?
- Could they make the analysis of ROC clearer? They used each parameter as an input to create a ROC curve? Or they used the probability of logistic regression to create the ROC curve? How did they get p value in table 4?
Specific comments:
1. They need to do proof reading. Many sentences does not make any sense and lots of grammar mistakes in the manuscript.
2. Give full name of ICU in the abstract.
3. Re-write this sentence in the abstract: ‘Treatment of increased ICP with mannitol or hypertonic saline solutions may cause fatal uncontrolled increase and lability of sodium, chloride and plasma osmolality values jeopardizing the patients’ outcome.’
4. Line 38, page 1: developing? I think it should be development?
5. Line 57-58, page 2, re-write this sentence please. ‘Variety of allowed and accepted proceedings included within osmotherapy impedes the comparability of the studies already focused on the issue.
6. Line 66, add the location details of the university hospital (city, country).
7. Line 103, page 3, add the location details of the Blood Gas System Analyser (city, country).
8. Line 203, page 6: established? Assumed or hypothesized?
Author Response
Dear Editors,
Thank you very much for this kind possibility to submit corrections of our paper to the Journal. We made all efforts to improve the manuscript according to the Reviewers’ suggestions. Below please find the answers to all queries raised in revision.
Kind regards, the Authors.
Reviewer #1
We appreciate your detailed assessment of our paper.
- 1. Your comment is accurate. In our study, we measured the ICP surrogate in the form of the optic nerve sheath diameter. To explain it to the Reader we have now described it with more details in Methodology and Limitations. We adjusted the paper title for it to better reflect our study as well. (lines 2-4, 112-137, 329-336)
- 2. We routinely did arterial blood gas tests twice a day in 12-hour intervals. We corrected the Methodology section to specify this. (lines 138-139)
- 3. Demographic factors put into the logistic regression model were age and gender, as depicted in the Table 5 and Methodology. (line 164-165)
- 4. In the ROC analysis we used each parameter as an input to create a ROC curve in three subgroups. The p values in Table 4 are the p values of each consecutive ROC curve. We corrected the Methodology section to specify this. (lines 168-169)
Specific comments:
- We corrected the manuscript. It was submitted for proof-reading by a native speaker.
- 2. We corrected the abstract and added a full name of the Intensive Care Unit (ICU). (lines 15-16)
- 3. We corrected the conclusion of the study to make it clearer and avoid misunderstandings, therefore, the sentence was rewritten. (lines 28-32)
- 4. We corrected the line according to the suggestion. (line 40)
- 5. We rewrote the sentence. (62-64)
- 6. We added the location details. (lines 71-72)
- 7. We added the location details. (line 139)
- 8. We corrected the sentence according to the suggestion. (line 246)

Reviewer 2 Report
I have carefully reviewed the manuscript entitled "Old question-new answer: is hyperosmolar therapy the best treatment of increased intracranial pressure? A single-center cohort study," and I regret to inform the authors that the paper in its present form is not suitable for publication in the journal. Firstly, this paper should be published as a letter to the editor or a short communication at most, as it does not fulfill the criteria of an original article. Secondly, there are some critical concerns regarding the quality of the paper that need to be addressed by the authors, including:
1. the title needs to be changed, as it may be misleading - results obtained from observational cohort studies do not warrant such statements
2. Although the clinical implication of the study does not raise any concerns, the authors failed to define the study rationale clearly
3. The reason for choosing one of the three treatment regimens was not clearly defined - there is an extremely high risk of selection bias
4. The ICP measurement was mentioned only once in the limitations section of the paper, and the chosen method is still not validated, which raises an important question about the applied methodology of the study
5. The entire material and methods section should be improved, as it does not provide sufficient data on the study protocol
6. The results do not support the conclusions - it appears that these are mostly overstatements
Author Response
Dear Editors,
Thank you very much for this kind possibility to submit corrections of our paper to the Journal. We made all efforts to improve the manuscript according to the Reviewers’ suggestions. Below please find the answers to all queries raised in revision.
Kind regards, the Authors.
Reviewer #2
We appreciate your inquisitive review of our paper. We hope our responses will rise to your expectations regarding applied corrections.
We find your comment regarding the fulfilment of the criteria for an original paper somewhat hurtful. The study was set up as an original study. The project was then approved by the local Ethics Committee. The prepared paper fulfils the criteria of an observational cohort study following the STROBE checklist and was reported as such. Please find below a detailed STROBE checklist with references to its particular elements in our manuscript to confirm this explanation.
- We slightly modified the title. But please remember that we phrased the title as a question not a statement, to focus the attention and interest of the Reader and to soften the conclusions in terms of causality. Also, we included the type of the study in the title to avoid any misunderstandings. We are however aware of the fact that an interventional randomized clinical trial would be a reference study to verify our findings. (lines 2-4)
- The study rationale was clearly defined in lines: 43-45, 50-51 and 62-64.
- The detailed description of the treatment selection process was included in lines 102-114. However, although the therapy was led according to Standard Operating Procedure (lines 102, 103) we cannot rule out the selection bias completely. To come up to your expectations, we included this in the study limitations. (lines 321-323)
- We expanded upon the method of ONSD measurements in the Methodology and Limitations sections, adding the detailed description method supported by already published articles. We selected a non-invasive method of ICP monitoring, which is not without risk of false measurement, however, there are studies supporting its use on a regular basis, and in our study the measurements were taken only by trained specialists with experience in utilizing this method in their clinical practise. (lines 114-137, 329-336)
- We applied corrections to the material and methods section. All STROBE criteria were included in the paper. Study group selection was shown in Figure 1.
STROBE guidelines has been used for article preparation (see: line ). All items are covered in the paper.
|
Item |
Line |
Item |
Line |
Item |
Line |
Item |
Line |
Item |
Line |
|
1 |
2-4, 11-32 |
6 |
72-82, Figure 1, 150-151 |
11 |
155-160 |
16 |
173-241 |
21 |
341-342 |
|
2 |
43-45, 50-51, 62-64 |
7 |
87-88, 95-101, 138-142 |
12 |
154-171 |
17 |
173-241 |
22 |
354 |
|
3 |
65-68 |
8 |
126-142 |
13 |
72-82, Figure 1, |
18 |
243-247, 280-285 |
|
|
|
4 |
70-81 |
9 |
161-166, 317-336 |
14 |
70-81, 87-88, 150-151 |
19 |
317-336 |
|
|
|
5 |
70-81, 150-151 |
10 |
81-82 |
15 |
199 |
20 |
338-342 |
|
|
- We rephrased the conclusions section as suggested to make our message clearer. (lines 338-342)

Reviewer 3 Report
This is an interesting article concerning the use of osmoactive agents for the treatment of intracranial hypertension, evaluating their efficacy performing different analyses on the blood. The article is well-structured, giving a complete insight of this challenging topic.
There are minor English grammatical errors in the text, that should be revised.
Some acronyms are not specified (e.g. CNS in the abstract)
One aspect that should be clarified and discussed regards the optic nerve sheath diameter measurement. The authors stated that “the diagnosis of intracranial hypertension was primarily based on clinical assessment and the ultrasonic measurement of an optic nerve sheath diameter was the only method routinely used in our patients”, without describing how this technique was performed and which criteria were utilized to make the diagnosis. Even if the goal of this study is focused on the intracranial hypertension treatment, this methodological aspect, with some related references (e.g. De Bernardo M, et al. Optic nerve ultrasound measurement in
multiple sclerosis. Acta Neurol Scand. 2019;139(4):399‐400) should be discussed in the “Study limitations” section or, even better, in the “Materials and Methods” section, because it can affect the patients’ selection. In fact, if B-scan ultrasonography was performed, this would represent a further limitation due to its pitfalls in measuring optic nerve sheath diameter.
Author Response
Dear Editors,
Thank you very much for this kind possibility to submit corrections of our paper to the Journal. We made all efforts to improve the manuscript according to the Reviewers’ suggestions. Below please find the answers to all queries raised in revision.
Kind regards, the Authors.
Reviewer #3
We would like to thank you for the detailed review and overall enthusiasm towards our study and its position as a challenge for the intensivists. Below please find the responses to issues raised in your review.
- We revised the text in the context of its grammatical correctness and improved certain misspelled words and other grammatical and contextual errors.
- We specified some acronyms that were previously not explained in their full-name form. (lines 12-13, 15-16, 18)
- According to your suggestions, we described in more details the technique of optic nerve sheath diameter measurements in the Methodology and Limitations sections. We added some necessary references as well. (lines 112-137, 326-336)

Round 2
Reviewer 1 Report
I recommend to accept the manuscript.
Author Response
We would like to thank you kindly for your recommendation to accept our manuscript for publication. We are glad our previous corrections met your expectations.
Kind regards, the Authors.

Reviewer 2 Report
First of all, I would like to thank the authors for addressing my concerns regarding the quality of their paper and their attempts to correct some of the major points I made. I need to admit that the quality of the manuscript has dramatically improved, yet I still have identified some potential flaws that need to be clarified:
1. I still find the title misleading. Since all the patients received some form of hyperosmolar therapy due to ICH, the question included in the title cannot be answered.
2. The authors included many unanswered questions regarding the clinical utility of hyperosmolar therapy in the introduction section, yet their study fails to address most of them.
3. The rationale for choosing the specific hyperosmolar therapy has not been explained in detail. Firstly, it's still impossible to tell whether the specific therapy was chosen according to the local protocol or based on the clinician's discretion. Secondly, the authors failed to provide any data regarding the initial value of ICP, therefore it is possible that some of the patients received hyperosmolar therapy without definite indications (e.g. for ICH prophylaxis as mentioned in the paper).
Author Response
Dear Dr. Czuczwar,
Thank you for your valuable comments. We do hope that our explanations will come up into your expectations and the paper will be ready for publication now.
- We modified the title into: “Hyperosmolar treatment for patients at risk for increased intracranial pressure: a single-centre cohort study”.
- We have intentionally posted those unanswered questions regarding the clinical utility of hyperosmolar therapy in “Introduction”. This section should familiarize the Reader with current knowledge of the subject and show where there are clinical and research uncertainties. Having read this part of the article, the Reader would know that there is a paucity of data regarding this topic and there is some space for further investigations. This is usually called the study rationale. Such an extensive approach to the problem is the result of giving binding answers to your previous review.
- We explained the rationale for choosing the specific hyperosmolar therapy in detail, as shown in lines 102-111. We applied the rules described in our hospital SOP document. Osmotic therapy was personalized and patient-oriented. We added this information (lines 109-113).
- We do not agree with your comment that we failed to provide any data regarding the initial value of ICP. We applied osmotic therapy based on repeatable ONSD measurements, which was described in detail in lines 114-139. ONSD was considered a surrogate of ICP and the treatment was applied based on the cut off value of 5.7mm (lines 136-137).
Kind regards, the Authors.
